

# Self-reported interoceptive accuracy and interoceptive attention differentially correspond to measures of visual attention and self-regard

Erik M. Benau

Psychology, State University of New York at Old Westbury, Old Westbury, NY,
United States of America

Corresponding author
Erik M. Benau,
benaue@oldwestbury.edu

## ABSTRACT

**Background**. Interoception, the perception of bodily functions and sensations, is a crucial contributor to cognition, emotion, and well-being. However, the relationship between these three processes is not well understood. Further, it is increasingly clear that dimensions of interoception differentially corresponds to these processes, yet this is only recently being explored. The present study addresses two important questions: Are subjective interoceptive accuracy and interoceptive attention related to self-regard and well-being? And are they related to exteroceptive (visual) attention?

**Methods**. Participants ($N = 98$; 29% women; aged 23–64 years) completed: a battery of questionnaires to assess subjective accuracy (how well one predicts bodily sensations), interoceptive attention (a tendency to notice bodily signals), self-regard (self-esteem, self-image, life satisfaction), state negative affect (depression, anxiety, and stress), a self-esteem Implicit Association Task (a measure of implicit self-esteem), and a flanker task to assess visual selective attention. Subjective interoceptive accuracy and attention served as dependent variables. Correlations and principal component analysis was used to establish correlations among variables and determine how, or whether, these measures are associated with subjective interoceptive accuracy or attention.

**Results**. Greater scores on measures of self-regard, implicit self-esteem, cognition and lower negative affect were broadly associated with greater subjective interoceptive *accuracy*. Conversely, only explicit self-esteem, satisfaction with life, and self-image corresponded to subjective interoceptive *attention*. An exploratory analysis with a more inclusive scale of interoceptive attention was conducted. Results of this exploratory analysis showed that the broader measure was a stronger correlate to self-regard than subjective interoceptive accuracy, though it, too, did not correlate with visual attention. In short, both subjective interoceptive accuracy and attention corresponded to well-being and mental health, but only accuracy was associated with exteroceptive attention.

**Conclusion**. These results add to a growing literature suggesting different dimensions of (subjective) interoception differentially correspond to indices of well-being. The links between exteroceptive and interoceptive attention, and their association with merit further study.

## INTRODUCTION

Interoception is the perception of one's bodily functions and sensations (*Tsakiris & Critchley, 2016*). Interoception is observed along three mostly independent dimensions: interoceptive accuracy (IA; performance on a behavioral measure of interoception); interoceptive sensibility (self-reported interoception *via* questionnaire), and interoceptive awareness (self-reported confidence in one's performance on an objective measure or accurate beliefs about one's ability to detect bodily signals in one context *vs.* another) (*Garfinkel et al., 2016*). Put another way, some people may be able to accurately detect their bodily signals when instructed to do so in a lab (typically detecting one's heartbeat), but may not be able to accurately perceive bodily signals *in vivo*; others may be aware that they are poor at detecting these signals on a daily basis, but may be able to do so when instructed in a lab, and so on (*Murphy et al., 2020*; *Murphy, Catmur & Bird, 2019*). For example, a recent study of female undergraduates showed no longitudinal associations between body surveillance and accuracy on a lab-based objective measure of IA (*Drew et al., 2020*). In a separate study, individuals with obsessive compulsive disorder showed little or no correspondence between interoceptive awareness and their performance on an objective measure of interoceptive accuracy, though control groups did show such correspondence (*Yoris et al., 2017*). Better performance on objective measures of interoception (*e.g.*, heartbeat detection) have generally corresponded to improved cognitive and emotional processing (*Suksasilp & Garfinkel, 2022*), whereas similar associations with interoceptive sensibility and awareness have been mixed (*Murphy, Catmur & Bird, 2019*). The inconsistent relationships of cognition and emotion to interoceptive sensibility may be due to differences in measurement (*Murphy, Catmur & Bird, 2019*; *Suksasilp & Garfinkel, 2022*).

### Self-reported interoceptive accuracy *vs.* interoceptive attention

All inventories putatively measure the lived experience of interoception, yet the most common interoception-related questionnaires have been found to index discreet dimensions (*Desmedt et al., 2022*; *Vig, Koteles & Ferentzi, 2022*). Most available questionnaires focus on *attention* and emotional reactions to bodily sensations (*e.g.*, noticing sneezes or being distressed or distracted by them) and comparatively few have focused on *accuracy* of bodily signals (*e.g.*, accurately predicting sneezes) (*Desmedt et al., 2022*; *Murphy, Catmur & Bird, 2019*). Murphy and colleagues (*2020*; *2019*) suggest that a person may pay significant attention to their bodily signals, yet this attention does not necessarily result in accurate perception or prediction of these signals. For example, individuals may report habitual attention to bodily signals but still report difficulties knowing when they are hungry, thirsty, or satiated (*e.g.*, *Fillon et al., 2021*; *Stevenson, Mahmut & Rooney, 2015*). Nevertheless, self-reported interoceptive *accuracy* has shown promise as a correlate to well-being and mental health like those of objective/physiological measures (*e.g.*, *Brand, Petzke & Witthöft, 2022*; *Trevisan et al., 2019*; *Ventura-Bort, Wendt & Weymar, 2021*). Only one study to date has compared the relationship between subjective interoceptive *accuracy* and interoceptive *attention* to measures of well-being and mental health, and found that interoceptive attention was broadly associated with symptoms across

psychopathologies whereas accuracy was specifically associated to internalizing symptoms (*Brand, Petzke & Witthöft, 2022*). Given the novelty of this line of research, it remains unclear what aspects of mental health and well-being *attention* and *accuracy* similarly correspond and where they may diverge.

## Interoception, self-regard, and well-being

IA and interoceptive sensibility—mostly interoceptive attention—are associated with improvements in social competence, resilience, and mental health, though the mechanisms underlying this association are not yet clear (*Baiano et al., 2021*; *Brand, Petzke & Witthöft, 2022*; *Eggart et al., 2019*; *Tsakiris & Critchley, 2016*). A prevailing explanation is that interoception enhances the ability to distinguish "self" from "other," resulting in improved, accurate self-representation without input from external resources or other people (*Frewen et al., 2020*). In other words, interoception (self-reported and otherwise) corresponds to the ability to detect self-referential stimuli (*Frewen et al., 2020*; *Garfinkel et al., 2013*). This self-referential ability then facilitates activating psychosocial processes that reduce distress and promote well-being (*Baiano et al., 2021*; *Garfinkel et al., 2016*). However, a simple question has yet to be asked: is the "self" at the center of that noise a valued entity? In other words, do IA and interoceptive sensibility correspond to *liking* and valuing oneself? It is well-established that interoception and depressive symptoms are negatively associated (*Eggart et al., 2019*), indicating that IA may be yoked to self-referential processing. However, depression and *self-regard* (an umbrella term that includes self-esteem and other self-evaluations) are independent constructs (*Orth & Robins, 2022*): low self-esteem conveys risk for depression, yet it is certainly possible to believe oneself to have value and still be depressed (*Yang et al., 2023*). It is surprising that no study to date has established if any measure of interoception is associated with and self-regard. This gap is important to address as self-regard and self-esteem are critical predictors of well-being and mental health in the general population (*Orth & Robins, 2022*).

## Interoceptive sensibility, exteroceptive attention, and well-being

There is increasing evidence that higher-level cognition, including executive functioning and attentional ability, is associated with mental and physical health (*Diamond, 2013*; *Gray-Burrows et al., 2019*). Exteroceptive attention is also positively linked to self-regard and well-being (*Gyurak et al., 2012*; *Pulopulos et al., 2022*). Whereas IA may facilitate grounding and ability to identify the "self" as a signal (*Frewen et al., 2020*), exteroceptive attention (and cognitive control more broadly) may enable an individual to disengage with aversive stimuli to help regulate mood (*Liu et al., 2019*). Improved attentional abilities may also facilitate downregulation to delay gratification, reduce impulsivity, and allow more judicious decision making (*Doidge, Flora & Toplak, 2021*; *Tan et al., 2023*).

There are increasing calls to understand the link between interoception, exteroception, and/or well-being (including emotional processes) as these are all typically disrupted across clinical populations, including autism (*Proff et al., 2022*), depression and anxiety (*Paulus & Stein, 2010*), and eating disorders (*Cusack et al., 2022*; *Herbert, 2020*), among others (*Bonaz et al., 2021*; *Khalsa et al., 2018*; *Owens et al., 2018*). Despite these calls, few researchers have

endeavored to establish these associations. At a practical level, there may simply be some overlap between the ability to avoid distraction and still identify one's bodily signals (*Buldeo, 2015*), and establishing these boundaries can clarify the form and function of attention. Of the limited available studies that have probed the link between subjective interoception and exteroceptive cognition, the results have been mixed: some studies have shown a positive relationship between interoceptive and exteroceptive (visual) attention (*Rae et al., 2020*) whereas others showed no relationship (*Rae et al., 2018*; *Vig, Ferentzi & Koteles, 2021*). All self-report measures assessed interoceptive attention. No other study included measures of well-being or emotion, though *Rae et al. (2020)* measured impulsivity and found no association with interoceptive attention. *Schultchen et al. (2020)* found that visual attention and objectively measured IA were independently associated with depression (the authors did not include self-report measures of interoception). *Haustein et al. (2023)* found similar patterns of associations in a sample of older adults. No other study has included measures of interoception, exteroceptive/visual attention, and mental health. Thus, it remains relatively unknown whether, and how, interoception (self-reported or otherwise) and exteroceptive attention contribute to well-being independently or in combination.

## Current study

Indices of exteroceptive attention, self-regard, and interoception are emerging as important correlates to well-being and mental health. However, no study to date has established if these variables do, in fact, correspond and to what degree they are associated with subjective interoception. Therefore, the goals of the present were threefold: (a) to establish whether indices of self-regard, well-being, and visual attention correspond to each other and (b) to determine whether these measures correspond to subjective interoceptive accuracy and/or (c) interoceptive attention. To achieve these goals, 98 adults completed a battery of questionnaires to measure subjective interoceptive accuracy, interoceptive attention, (positive) self-regard, and negative affect. Participants also completed an implicit self-esteem task and a selective attention task. It was hypothesized that measures of negative affect, self-regard, and attention would correlate with each other and, in turn, load onto a single component within a principal component analysis. Scores on this component would then be positively associated with both subjective interoceptive accuracy and interoceptive attention. More specifically, both interoceptive measures would be associated with lower scores on negative affect and greater scores on all other measures. Given the novelty of the question, an exploratory goal was to determine which measure (accuracy *vs.* attention), if either, better corresponds self-regard and well-being.

## METHODS

### Participants

Participants were recruited *via* Amazon's Mechanical Turk (MTurk) service, a platform for obtaining crowd-sourced data. Only participants who registered their location in the United States were able to participate in the study. A total of 184 responses were collected. Six participants completed the study twice and only their first responses were included in analyses. An additional 80 participants did not complete one or both behavioral tasks

and were removed from analyses, leaving 98 responses ($n = 28$ [29%] women), or about 53% of the original sample. It is unclear why there was such a high rate of attrition (see discussion section). Little's test indicated that the data were not missing completely at random $\chi^2$ (109) = 146.12, $p = .010$. Completers and non-completers were statistically similar on demographics (age, gender, income, and education level; $ps > .20$), implicit and explicit self-esteem, self-image, IAS, SWLS, MAIA noticing or MAIA-g scores ($ps > .057$, $ds < .29$, these measures are described below). Compared to completers, non-completers reported significantly greater depression, anxiety, and stress ($ps < .001$; $d = .77$–.85). Given the mix of these results and the absence of systematic missingness, the present dataset can be diagnosed as having data *missing at random* (*Mack, Su & Westrich, 2018*).

All included participants answered every question and passed at least four of five attention checks (described below). Included participants were aged 23–64 years ($M = 36.21$, SD $= 10.20$). Most (78%) of the sample identified as White and non-Hispanic (74%). Most (88%) of the sample reported having completed at least a bachelor's degree. Modal yearly household income (37%) was between $50,000-$75,000; 66% of the sample reported annually making $75,000 or less and 34% reported making $100,000 or more. One respondent refused to provide their income.

There is no established protocol to determine power for principal component analysis (PCA) as conducted in this study. *A priori* power analyses using G*Power software (*Faul et al., 2009*; *Faul et al., 2007*) indicated that at least 84 participants were needed to adequately power correlation analyses with a medium effect size ($1-\beta = .80$; $|r| \geq .3$); the final sample exceeded that minimum. This effect size is smaller than that presented by *Rae et al. (2020)*, $r = .438$ ($N = 43$), which could easily be powered by the current sample of 98 ($1-\beta = .99$). Nevertheless, some argue that much larger samples (*e.g.*, over 250 responses) are necessary for stable correlational analyses (*e.g.*, *Schönbrodt & Perugini, 2013*). Due to budgetary and logistical constraints, a larger sample was not feasible to attain, let alone one exceeding 250 respondents. Therefore, even though this sample met *a priori* power analyses for moderate correlations, the results here should be considered preliminary and additional studies are necessary to confirm these findings.

### Questionnaires

Scales and measures below were either (or a combination) expressly available in the public domain, permitted by the authors of the instruments in either a blanket permission or specifically obtained by the research team, included in publications implying permission to be used, or a license was duly purchased. Details regarding copyright are available upon request.

*Self-Regard*. Self-esteem, a view of one's worth, was measured using the 10-item Rosenberg Self Esteem Scale (RSES; *Rosenberg, 1965*). Questions assess both positive and negative feelings about the self (*e.g.*, "*I feel that I have a number of good qualities.*") that are answered on a four-point scale ranging from 0 (*strongly disagree*) to 3 (*strongly agree*) where higher scores indicate greater self-esteem. This sample demonstrated good internal consistency on the RSES ($\alpha = .86$). Whereas self-esteem is a broad (positive) sense of self, *self-image* is an evaluation of specific traits that reflect how one believes they are viewed

by others (*Bailey 2nd, 2003*). To measure self-image, participants completed the 20-item Flush Self-Image Scale (FSIS; *Reynolds, 2002*). The FSIS is a semantic differential scale where each item contains a pair of adjectives and respondents move a slider on a continuous scale toward whichever item in the pair they believed described them better. All negative items were anchored at 0 and positive items were anchored at 7 (*e.g.*, 0 ["*ugly*"] to 7 ["*beautiful*"]); order of positive and negative items was pseudo-randomized and presented in the order of the original manuscript. Numeric values of responses were not shown to participants. Mean distance toward positive items (rounded to one decimal) was used in all analyses: higher scores indicate higher self-image. Internal consistency on the FSIS was excellent ($\alpha = .96$). Finally, the five-item *Satisfaction with Life Scale* (SWLS; *Diener et al., 1985*) was included to measure a respondents overall subjective well-being. The SWLS includes personal satisfaction as a key component that is broadly defined without emotional content (*e.g.*, "*I am satisfied with my life*"). Responses were measured on a Likert-type scale ranging from 1 (*strongly disagree*) to 7 (*strongly agree*). Internal consistency on the SWLS was excellent ($\alpha = .90$).

*Interoception.* Self-reported interoceptive accuracy was measured using the 21-item Interoceptive Accuracy Scale (IAS; *Murphy et al., 2020*). The IAS is the only available instrument that specifically measures the dimension of accuracy, and not attention, within interoceptive sensibility. The IAS asks if the respondent can "always accurately perceive" 21 specific bodily functions or sensations (*e.g.*, "*I can always accurately perceive when I am thirsty*"). Questions on the IAS are answered on a scale of 1 (*strongly disagree*) to 5 (*strongly agree*), where higher scores indicate greater perceived interoceptive *accuracy*. The IAS is one of relatively few self-report measures of interoception that have correlated with objective measures (*e.g.*, *Murphy et al., 2020*). The present sample showed good internal consistency on the instrument ($\alpha = .87$).

Interoceptive attention was measured using the Multidimensional Assessment of Interoceptive Awareness (MAIA; *Mehling et al., 2012*). The MAIA, a gold standard measure of interoceptive sensibility (*Desmedt et al., 2022*; *Vig, Koteles & Ferentzi, 2022*), contains 32 items inquiring about experiences of interoceptive attention and awareness across eight subscales: Noticing, Attention Regulation, Emotional Awareness, Self-Regulation, Body Listening, and Trusting subscales, Not Worrying, and Not Distracting. Items are measured on a Likert-type scale ranging from 0 (*never*) to 5 (*always*). The four-item "noticing" subscale was used since it measures "awareness of uncomfortable, comfortable, and neutral body sensations" (*Mehling et al., 2012*), which is most germane to the goals of the present study (*e.g.*, "*I notice when I am uncomfortable in my body*"). The subscale attained acceptable internal consistent ($\alpha = .67$). As an exploratory step, the "MAIA-g" scale was calculated by taking the average score of each scale except Not Worrying and Not Distracting (*Ferentzi et al., 2021*). Though this calculation somewhat defeats the purpose of the intended multidimensionality of the MAIA (*Mehling et al., 2012*), this common score has been found to correspond well with other gold standard measures of interoceptive attention (*Ferentzi et al., 2021*). The strengths and limitations of this approach are considered in the discussion section.

*Negative Affect*. The 21-item Depression, Anxiety, and Stress Scale (DASS-21; *Henry & Crawford, 2005*) measures past-week negative affect across three dimensions with eponymous subscales (*Depression*, *Anxiety*, and *Stress*) on a scale of 0 ("*Did not apply to me at all*") to 4 ("*Applied to me very much, or most of the time*"). Sample items include: "*I felt that life was meaningless*" (depression), "*I felt scared without any good reason*" (anxiety), and "*I found it hard to wind down*" (stress). The scale is well-established as sensitive to negative affect in non-clinical samples (*e.g., Henry & Crawford, 2005*). Each subscale attained excellent internal consistency (each $\alpha$ >.93).

## Behavioral measures

*Implicit Self-Esteem*. The Self-Esteem Implicit Association Task, (IAT; *Greenwald et al., 2002*), was used to measure implicit self-esteem. In this task, participants are asked to categorize pleasant (*e.g.*, "Joy") *vs.* unpleasant (*e.g.*, "Filth") words paired with stimuli related to self ("Me") or other ("Others"). According to theory, greater congruence between concepts results in faster reaction time (RT) than incongruent pairs. Thus, if an individual responds faster to "self + pleasant" than "self + bad" words, this would indicate more immediate associations of the self as a pleasant category. With the exception of minor changes to instructions to match the goals of this study, the default settings from Millisecond software, including stimuli, timing, and other parameters of the IAT, matched those of a standard, counterbalanced administration (see: *Greenwald et al., 2002*).[1] The IAT was chosen to complement questionnaires as they are less susceptible to impression management (*i.e.,* harder to "fake") than self-report measures as it (*Rohner, Schroder-Abe & Schutz, 2011*). *D*-scores, the dependent variable of the IAT, are a calculated and weighted difference of RT to ("self + good" and "other + bad") from ("other + pleasant" and "self + bad") words. More positive *D*-scores indicate greater implicit self-esteem, and more negative scores indicate lower self-esteem. It should be noted that there is both long-standing support for the use of implicit measures to assess self-esteem that is not captured through self-report measures (*Hofmann et al., 2005*; *Pietschnig et al., 2018*), and long-standing controversy about the IAT and implicit attitude measurement more broadly (*Machery, 2022*; *Schimmack, 2021*). It was beyond the scope of this paper to delve into these controversies, but the extant evidence is sufficient to believe that implicit and explicit self-esteem index distinct facets of self-regard (*Hofmann et al., 2005*; *Pietschnig et al., 2018*).

*Selective Attention*. Participants completed a flanker task using the original parameters reported by *Eriksen & Eriksen (1974)* using the default settings in the Millisecond library (with minor wording changes to instructions).[2] Briefly, in this task, participants identified if the center letter in a string of letters was angular (H or K) or curvy (C or S). They pressed "P" on their keyboard if it belonged to one category or "Q" if it belonged to another (starting placement of the categories was randomly determined and the category placement changed midway through the experiment). Participants completed 648 trials. Targets were presented with no flankers or one of five flanker conditions (same letter, same category different letter, same category mixed letters, different category mixed letters). Trials were an even distribution of three spacings (no space between flankers, narrow spacing, wide spacing) and five flanker (noise) conditions for a total of 504 trials with

[1] More information here: https://www. millisecond.com/download/library/v6/ iat/selfesteemiat/selfesteemiat/selfesteemiat/ selfesteemiat.manual.

[2] More information here: https://www. millisecond.com/download/library/v6/ flankertask/flankertest_eriksen/flankertest_ eriksen/flankertask.manual

flankers and 72 trials without flankers across six "mixed" blocks. An additional six blocks of 12 trials with no flankers (three trials per target letter) were also presented as a control condition. All targets appeared above a fixation cross, and participants had 1 s to respond. Instructions were repeated every block.

Data from trials with no flankers within mixed blocks were not analyzed in the present study. RT and accuracy from this task were assessed in two ways. First, overall mean RT and accuracy gave an index of general performance on the task. Second, difference scores were calculated to determine to what degree the flanker trials in mixed blocks were distracting. The average accuracy and RT on trials in blocks with no flankers were subtracted from the average accuracy and RT on trials that had flankers (*i.e.,* $\bar{x}$ Accuracy$_{\text{flanker}}$ − $\bar{x}$ Accuracy$_{\text{NoFlanker}}$; $\bar{x}$ RT$_{\text{flanker}}$ − $\bar{x}$ RT$_{\text{NoFlanker}}$). More positive scores on accuracy indicate improved performance on flanker than non-flanker trials; more positive scores on RT indicate slower responses on trials with flankers than without.

**Procedures.** Participants completed consent procedures and questionnaires using Qualtrics software (Qualtrics, Inc., Provo, UT, USA). After questionnaires, participants clicked a URL that automatically downloaded the Inquisit player, the software that presented the IAT and flanker (Millisecond Software, Seattle, WA, USA). Order of IAT and flanker was randomized. After, participants returned to Qualtrics where they completed two quality-control questions assessing honesty.[3] Median completion time was 39 min. All included participants completed the study on a computer or tablet. In line with MTurk policies, participants were reimbursed $4.50 only if they completed all questionnaires and both behavioral tasks. The Institutional Review Board at SUNY Old Westbury approved the procedures in the current study under exemption category two and, as such, the protocol was not assigned an approval number to present here.

**Data analysis.** First, Pearson correlations were conducted across all variables of interest. Next, a PCA (and subsequent rotated PCA) was conducted with each variable of interest (except the IAS and MAIA noticing) was used to reduce the number of variables being assessed and determine their association with IAS and MAIA scores. PCA was chosen over factor analysis (and similar approaches) as the latter assumes an unmeasured latent variable, to some degree, contributes to the scores of the variables of interest. PCA is agnostic to cause and instead assesses correlations among measures to reduce the number of variables in analyses by generating component scores that go on to be analyzed. Component scores are standardized linear combinations of each variable's values within the component. It was hypothesized that all independent variables would load onto a single component based on the overlap between affect and cognition discussed above and in turn a potential association to interoception (*e.g., Bonaz et al., 2021; Khalsa et al., 2018; Owens et al., 2018*). However, the best fit for the data was a varimax-rotated PCA solution with two components (the process is described further below). Component scores were then (a) correlated with IAS and MAIA-noticing scores and (b) entered as independent variables in a multivariate regression model with IAS and MAIA noticing scales entered as the dependent variable(s). Multivariate regression calculates identical individual coefficients and standard errors as would be produced by running each model separately while estimating between-equation covariances to test coefficients across equations. Doing so allows a direct comparison of

[3]Participants were asked to rate their honesty (without penalty) on a continuous sliding scale from 0 ("not honest at all") to 10 ("completely honest") with no other anchors. One respondent placed the slider at 2.60 units to the first question; 40% selected 5.3−9.9; 60% selected 10. Participants also answered whether their data should be included in analyses. Two respondents selected "I don't know" and the rest reported "probably" or "definitely" yes. Two respondents did not answer this question but were retained in analyses.
coefficients to determine which dependent variable (in this case, IAS *vs.* MAIA noticing) is best predicted by a set of independent variables using a Wald *F*-test (*Stockemer, 2019*). In other words, multivariate regression allows for the ability to directly determine whether a set of independent variables predicts one dependent variable better than another dependent variable.

As an exploratory step, each analysis described above was also conducted with the MAIA-g swapped in for the MAIA noticing scale. The MAIA noticing and MAIA-g subscales were analyzed separately as the noticing subscale is part of the MAIA-g. There were no missing data in the present dataset. Data were analyzed using jamovi 2.3.21 (jamovi project, Sydney, Australia) and Stata 17.0 (Stata, Inc., College Station, TX, USA). Data are available here: https://osf.io/7sxe3/.

# RESULTS

Descriptive statistics of, and correlations between, all measures are shown in Table 1 (a full correlation matrix among all measures is presented in Appendix A). The IAS robustly correlated with nearly all self-regard measures, negative affect, implicit self-esteem, and visual attention (but not the difference scores for accuracy of flanker and no-flanker trials). Notably, the correlation between IAS scores and RT on flanker (*vs.* no-flanker) indicates that, as IAS scores increased, the odds of faster responses with flankers than without also increased (though accuracy was the same on both conditions).[4] Figure 1 presents the scatterplots of the correlations of the variables of interest to the IAS. Conversely, the MAIA noticing score only correlated with the overall accuracy of the flanker task and no other measure. Importantly, not all variables correlated with each other, indicating little or no acquiescence and/or bias to socially desirable responding (elaborated further in the discussion section).

## Principal components analysis

An initial PCA showed three components with eigenvalues greater than 1.0. Several items cross-loaded on components rendering them uninterpretable. To clarify interpretation, promax rotation (oblique/correlated) was used, but the correlations between component 2 to components 1 and 3 were weak ($|r| < .19$). Only the two measures of flanker RT loaded onto the third factor, which was highly correlated with the first ($r = -.453$). An orthogonal (uncorrelated) varimax rotation improved component interpretability, but the two measures of flanker task RT remained the sole variables on the third component. Forcing a two-component solution resulted in all variables loading onto one of the factors sufficiently (all factor loadings $\geq |.4|$). A unitary solution resulted in the RSES, FSIS, and SWLS not loading sufficiently on the component (loading $< |.4|$). Thus, the two-component structure provided the best fit and was the final, included component structure in this study (Table 2). The progression of the PCA solution is presented in Appendix B. The two components accounted for about 59% of the variance.

Component one consisted of all behavioral measures (flanker accuracy and RT, and IAT *D*-scores), which loaded positively, and the three subscales of the DASS, which loaded negatively. Component two consisted of explicit self-esteem (RSES), self-image (FSIS),

[4] Scoring above the median on the IAS corresponded to an 88% greater chance of improved accuracy on the flanker than no-flanker trials, OR = 1.875, 95% CI [0.83–4.23], indicating that the flankers may have been facilitative instead of distracting.

**Table 1 Descriptive statistics for each measure and their correlation (Pearson r) with IAS and MAIA.**

| | M (SD) | Correlation with IAS | Correlation with MAIA Noticing | Correlation with MAIA-g |
|---|---|---|---|---|
| IAS | 84.46 (10.94) | — | .539*** | .506*** |
| MAIA | 3.19 (0.81) | .539*** | — | .698*** |
| MAIA-g[a] | 3.29 (0.65) | .506*** | .698*** | — |
| RSES | 19.96 (2.17) | .266** | .068 | .161 |
| FSIS[b] | 4.61 (1.38) | .262** | .094 | .438*** |
| SWLS | 21.30 (7.51) | .316** | .108 | .331*** |
| DASS-D | 13.96 (6.20) | −.206* | −.191 | −.242* |
| DASS-A | 13.74 (6.15) | −.219* | −.167 | −.101 |
| DASS-S | 14.79 (6.00) | −.232* | −.143 | −.157 |
| IAT | 0.22 (0.51) | .232* | .181 | .252* |
| Flanker RT | 505.64 (231.70) | .310** | .259* | .143 |
| Flanker Acc | 0.72 (0.24) | .213* | .034 | .021 |
| Flanker RT (diff) | 33.06 (318.14) | .128 | −.037 | −.051 |
| Flanker Acc (diff) | −0.02 (0.32) | .222* | .136 | .095 |

Notes.

IAS, Interoception Awareness Scale; MAIA, Multidimensional Assessment of Interoceptive Awareness–Noticing Subscale; MAIA-g, the "general" factor of the MAIA (see text); RSES, Rosenberg Self-Esteem Scale; FSIS, Flush Self-Image Scale; SWLS, Satisfaction With Life Scale; DASS-D, DASS-A, and DASS-S, Depression Anxiety and Stress Scale and same-named subscales; IAT, $D$-scores of the self-esteem Implicit Association Task; Flanker Overall RT, RT (ms)for the whole task; Flanker Overall Acc, Proportion of correct answers for the whole flanker task; Flanker Diff. RT, Difference of RT (ms) no-flanker blocks from flanker trials in mixed blocks; Flanker Diff. Acc., Difference of proportion of correct answers on no-flanker blocks from flanker trials in mixed blocks.

[a] The average response across six (of eight) subscales on the MAIA (see text).

[b] The mean distance from negative attribute to positive attribute on a scale of 0–7.

*$p < .05$.

**$p < .01$.

***$p < .001$.

and life satisfaction (SWLS). The first component seems to represent *signs and symptoms* of cognition and emotion, whereas the second component seems to represent explicit *self-regard*.

## Relation of principal component scores to self-reported interoceptive accuracy and interoceptive attention

Table 3 presents an overview of the following results. IAS scores significantly correlated with the *signs and symptoms*, $r = .244$, $p = .015$, and *self-regard* components, $r = .316$, $p = .002$. Conversely, MAIA noticing scores neither correlated with *signs and symptoms*, $r = .169$, $p = .097$, nor *self-regard*, $r = .105$, $p = .302$. Results of multivariate regression suggested that the IAS could be significantly predicted by the two component scores, $R^2 = .16$, $F (3,95) = 9.00$, $p < .001$. IAS scores were significantly predicted by the *self-regard*, $b = 3.45$ ($\beta = .32$), $SE_b = 1.03$, $t = 3.35$, $p = .001$, and the *signs and symptoms* components, $b = 2.67$ ($\beta = .24$), $SE_b = 1.03$, $t = 2.60$, $p = .011$. Conversely, the model did not suggest the MAIA noticing could be predicted by the two components, $R^2 = .04$, $F (3,95) = 1.95$, $p = .148$. Within this model, it was found that the MAIA noticing scale was neither predicted by scores on the *signs and symptoms* component, $b = 0.14$ ($\beta = .17$), $SE_b = 0.08$, $t = 1.68$, $p = .097$, nor the *self-regard* component, $b = 0.08$ ($\beta = .11$), $SE_b = 0.08$, $t =$

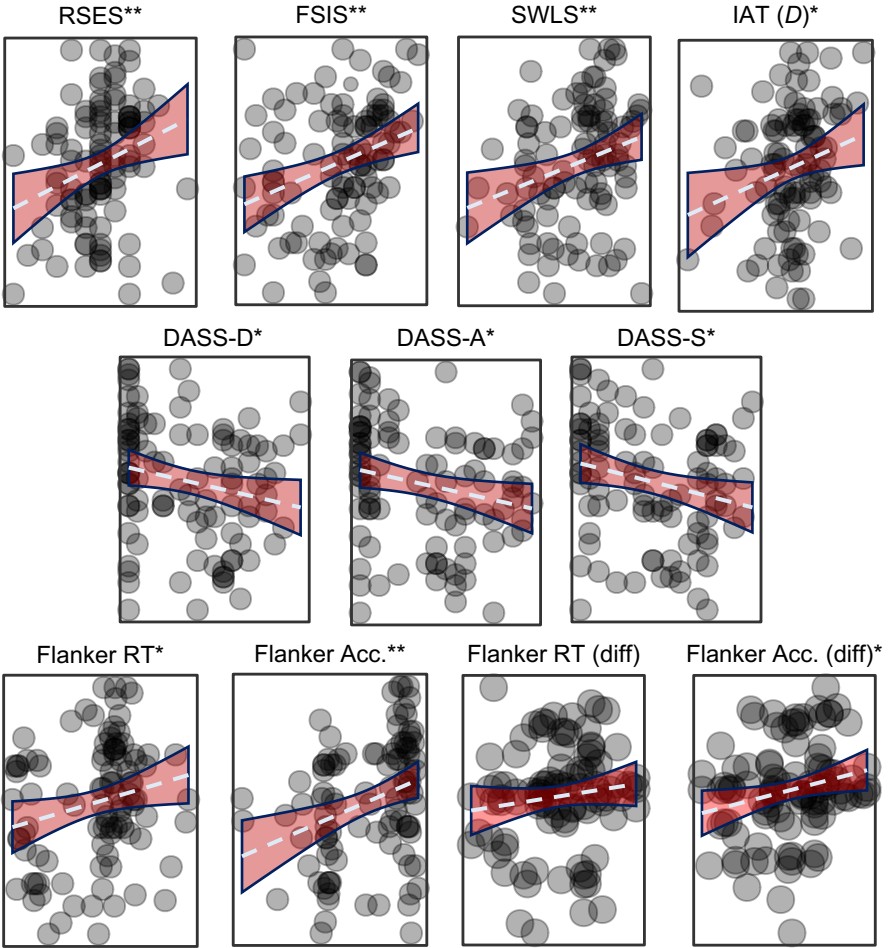

**Figure 1** **Scatterplots showing the association between the IAS and the measures of interest.** Measures of self-evaluation are in the top row, negative affect middle row, and measures of attention in the bottom row. The IAS is on the Y axis and the noted measure is on the X axis. Dashed blue line is the trendline, and red area indicates the 95% CI of the correlation. RSES, Rosenberg Self-Esteem Scale; FSIS, Flush Self-Image Scale; SWLS, Satisfaction With Life Scale; DASS-D, DASS-A, and DASS-S, Depression Anxiety and Stress Scale and same-named subscales; IAT (D), D-scores of the self-esteem Implicit Association Task; Flanker Overall RT, Reaction Time for the whole task; Flanker Overall Acc, Percent accuracy for the whole flanker task; Flanker Diff. RT, difference of RT no-flanker blocks from flanker trials in mixed blocks (no-flanker –flanker); Flanker Diff. Acc., Difference of percent accuracy no-flanker blocks from flanker trials in mixed blocks (no-flanker –flanker); *$p < .05$; ** $p < .01$.

1.05, $p = .297$. Finally, a Wald $F$-test indicated that the superiority of predicting IAS over MAIA noticing scores was not by chance, $F(4, 95) = 4.70$, $p = .002$, $d = 0.44$.

In sum, IAS scores corresponded with higher implicit and explicit self-regard, lower negative affect, and slower, more accurate responses on the flanker task. However, these same component scores were not statistically associated with the MAIA noticing subscale.

**Table 2** Final component structure of the PCA.

| | Factor 1 Scores (*Signs & Symptoms*) | Factor 2 Scores (*Self-Regard*) | Uniqueness |
|---|---|---|---|
| DASS-A | -.908 | | .148 |
| DASS-S | −.868 | | .197 |
| Flanker Acc | .841 | | .293 |
| DASS-D | −.836 | | .172 |
| Flanker Acc (diff) | .648 | | .551 |
| Flanker RT (diff) | .628 | | .604 |
| IAT-D | .566 | | .650 |
| Flanker RT | .560 | | .645 |
| SWLS | | .831 | .283 |
| FSIS | | .828 | .290 |
| RSES | | .425 | .731 |

Notes.
Varimax rotation was used: IAS, Interoception Awareness Scale; RSES, Rosenberg Self-Esteem Scale; FSIS, Flush Self-Image Scale; DASS-D, DASS-A, and DASS-S, Depression Anxiety and Stress Scale and same-named subscales; IAT-D, *D*-scores of the self-esteem Implicit Association Task; Flanker Overall RT, Reaction Time for the whole task; Flanker Overall Acc, Proportion of correct answers for the whole flanker task; Flanker Diff. RT, difference of RT no-flanker blocks from flanker trials in mixed blocks; Flanker Diff. Acc., Difference of proportion of correct answers on no-flanker blocks from flanker trials in mixed blocks.

**Table 3** Overview of correlation and regression results assessing the relationship of component scores to the IAS, MAIA noticing, and MAIA-g.

| IV | DV | Correlation result | Regression result |
|---|---|---|---|
| Signs and Symptoms | IAS[a] | Significant | Significant |
| | MAIA Noticing | NS | NS |
| Self-regard | IAS[a] | Significant | Significant |
| | MAIA Noticing | NS | NS |
| | *Exploratory Analyses with MAIA-g* | | |
| Signs and Symptoms | MAIA-g[b] | NS | NS |
| Self-regard | MAIA-g[b] | Significant | Significant |

Notes.
IV, Independent Variable; DV, Dependent Variable; Correlation result, independent association between component scores and DV; Regression result, whether the IV significantly predicted the DV when both components were entered into the model; IAS, Interoceptive Accuracy Scale; MAIA Noticing, Multidimensional Assessment of Interoceptive Awareness, "noticing" subscale; MAIA-g, General factor of the MAIA (see text); the bottom half of the table presents the associations with the MAIA-g (regression results of component scores predicting the IAS are identical to the top half); NS, not significant.
[a]The superiority of predicting the IAS over the MAIA-g in this model was not by chance ($p = .002$).
[b]The superiority of predicting the MAIA-g in this model was not by chance ($p < .001$).

## Exploratory analysis with the MAIA-g factor

The MAIA-g factor correlated highly with the IAS, $r = .506$, $p < .001$, and the *self-regard* component, $r = .426$, $p < .001$, but not the *signs and symptoms* component, $r = .098$, $p = .339$. Multivariate regression analyses indicated that the component scores significantly predicted the IAS (results of its regression model are identical to those above). The MAIA-g was also predicted by the two component scores, $R^2 = 0.19$, $F(3,95) = 11.23$, $p < .001$. Reflecting the correlation analyses, the MAIA-g was significantly predicted by the *self-regard*

component, $b = 0.277$ ($\beta = .426$), $SE_b = 0.92$, $p < .001$, but not the *signs and symptoms* component, $b = 0.06$ ($\beta = .100$), $SE_b = 0.09$, $p = .293$. Results of a Wald $F$-test suggests the superiority of the model predicting the MAIA-g over the IAS was not by chance, $F(4,95) = 7.62$, $p < .001$, $d = 0.56$.

## DISCUSSION

The goal of the present study was to establish whether, and how, self-reported interoceptive *accuracy* (measured by the IAS) and self-reported interoceptive *attention* (measured by the MAIA) correspond to indices of self-regard (implicit and explicit self-esteem, self-image, life satisfaction), negative affect, and visual attention. It was hypothesized that measures of self-regard and attention would correspond with each other (*i.e.,* load onto a single component) and, in turn, scores from this component would correspond to self-reported accuracy and self-reported attention. These hypotheses had mixed support. Instead of a single component, the independent variables loaded onto two: one that seemingly indexed objective signs and symptoms of cognition and emotion (implicit self-esteem, all measures of visual selective attention, and negative affect) and one that indexed subjective self-regard. Both component scores positively correlated with accuracy and independently predicted accuracy within a regression model. The results with interoceptive attention were more complex: MAIA "noticing" scores only correlated with accuracy on the flanker task and no other measure. However, an exploratory step of using the MAIA-g—the average of all but two subscales on the MAIA (*Ferentzi et al., 2021*)—resulted in stronger associations to self-regard than IAS scores, but was also statistically unassociated with visual attention.

The results here complement and extend previous work showing that interoceptive attention is associated with wellbeing (*e.g., Ferentzi, Horvath & Koteles, 2019*) and mental health (*Brand, Petzke & Witthöft, 2022*; *Eggart et al., 2019*; *Trevisan et al., 2019*). Further, the present results are among the first to find such associations with self-reported interoceptive *accuracy* (*e.g., Brand, Petzke & Witthöft, 2022*; *Ventura-Bort, Wendt & Weymar, 2021*), and the first to identify subjective interoceptive accuracy as a correlate to visual attention. These findings are discussed in turn below.

### Self-regard

Complementing and extending previous work (*Ferentzi, Horvath & Koteles, 2019*; *Schultchen et al., 2020*; *Ventura-Bort, Wendt & Weymar, 2021*), the results of this study suggest that greater scores on the IAS (subjective interoceptive accuracy) corresponded to reduced negative affect and greater self-regard. Thus, as hypothesized, greater subjective accuracy is associated with *liking* oneself, in addition to reduced negative affect (the latter is discussed further below). Whether subjective accuracy is cause or consequence of self-regard is unclear. It may be that (accurate) awareness of one's body and its reactions across contexts will facilitate setting oneself up for success, leading to a more satisfactory life, greater self-regard, and less negative affect. For example, life will almost certainly be more satisfactory for individuals who can accurately predict their digestive system (as inquired in the IAS) than those who cannot. Similar patterns likely apply for being able to predict injury, sexual arousal, and other bodily functions. It may also be that those with

positive self-regard and reduced distress are in a better position to monitor their body and its reactions across contexts. Indeed, initial longitudinal work provides support for the postulation that interoception predicts improved emotion regulation and well-being over time (*Tan et al., 2023*). It may also be that self-regard impedes accurate appraisal of accuracy: those high on self-regard may believe they are well in-tune with their body yet may not *actually* be better than anyone else (and the inverse may be true for lower-scoring individuals). Realistically, a combination of these explanations contributed to the findings observed here. Prospective work and/or objective measures are needed to confirm and clarify these relationships.

In contrast to subjective interoceptive accuracy, the relation of interoceptive attention to self-regard was complex. The MAIA noticing subscale did not correspond with self-regard, and it is not clear why. One possibility is the MAIA noticing subscale is not as robust or provides sufficient variance as would be needed to adequately capture interoceptive attention. For example, *Ferentzi, Horvath & Koteles (2019)*, who found significant associations between well-being and interoceptive attention, used the Body Awareness Questionnaire (*Shields, Mallory & Simon, 1989*), which contains 18 items, compared to the MAIA noticing with just four. However, *Ventura-Bort, Wendt & Weymar (2021)* also used several subscales of the MAIA (including the "noticing" subscale) in their analyses and identified a complex pattern of results. Therefore, it is not clear if the length and variance of the subscale itself can entirely account for the null findings in the present study. Using the "noticing" factor of the MAIA was motivated by the postulation that the scale is multidimensional and each factor in the instrument is able to be administered independently (*Mehling et al., 2012*). However, individual factor scores may not be the best use of this instrument, and several authors have recently noted the complexity of the MAIA (*Desmedt et al., 2022*; *Ferentzi et al., 2021*; *Vig, Koteles & Ferentzi, 2022*).

*Ferentzi et al. (2021)* suggest that the MAIA is not as multidimensional as assumed and, instead, the questionnaire can be better assessed along a general factor—the MAIA-g—an average of all but two problematic factor scores. Ferentzi and colleagues found that the MAIA-g provides better concurrent validity with other inventories than the MAIA noticing subscale alone and provides a more parsimonious measure to use in studies of interoception (*Ferentzi et al., 2021*; *Vig, Koteles & Ferentzi, 2022*). Following this suggestion, as an exploratory step in the present study, the MAIA-g was calculated and assessed within the same analysis plan as the noticing subscale. Results showed that the MAIA-g scale correlated with the FSIS, SWLS, though not the RSES. Despite no direct correlation with the RSES, the MAIA-g significantly correlated with *self-regard* component scores. In fact, multivariate regression models indicated that the *self-regard* component demonstrated superior associations to the MAIA-g over the IAS (though the IAS was also significantly predicted by the component). Interpretation of this result should be approached with caution as the current sample was not large enough to verify the unitary factor structure of the MAIA-g, nor have there been additional studies since Ferentzi and colleagues' (*2021*) that have done so. Nevertheless, these results provide further evidence that indices of mental health increase together with interoceptive attention and self-reported

accuracy (*Brand, Petzke & Witthöft, 2022*), though the precise pattern of associations will require further work.

### "Signs and symptoms" of emotion and cognition

Generally, greater subjective interoceptive accuracy corresponded to greater of visual attention, improved implicit self-esteem, and lower negative affect. This association was maintained when the scores were combined into a single component. The results associated with interoceptive attention were more mixed. The MAIA noticing subscale score only correlated with overall accuracy on the flanker task and did not correspond with implicit self-esteem or negative affect. Neither the MAIA noticing subscale nor the MAIA-g corresponded with the *signs and symptoms* component.

### Visual attention

Consistent with hypotheses, these results provide preliminary evidence that subjective interoceptive accuracy corresponds to visual attention. In fact, as subjective accuracy (IAS scores) increased, the likelihood of a positive difference score (indicating greater performance on flanker *vs.* no-flanker tasks) increased. In other words, the flankers potentially went from distracting to facilitative. This is the first study to measure selective visual attention whereas others measured inhibition (*Rae et al., 2020*; *Rae et al., 2018*) and/or sustained attention (*Vig, Ferentzi & Koteles, 2021*). Therefore, in line with the hypothesis that interoceptive accuracy facilitates finding the *self* as signal from noise (*Frewen et al., 2020*), it may be that it also corresponds with the ability to identify a *visual* signal from noise. In contrast, the MAIA noticing scale correlated to overall accuracy on the flanker whereas the MAIA-g did not correspond to any measure of visual attention. It may be that noticing bodily signals, specifically, corresponds to noticing external stimuli and this variance is washed out when including other aspects of interoceptive attention. It is difficult to argue that interoceptive attention, as measured in the present study, had a meaningful association with visual attention, especially when the MAIA-g had no associations at all. Previously, interoceptive attention has (*Rae et al., 2020*) and has not (*Rae et al., 2018*; *Vig, Ferentzi & Koteles, 2021*) correlated with performance on visual attention tasks. The present study provides little clarity on the association of interoceptive attention to visual attention and, if anything, adds more null findings to the literature. An important next step will be to confirm and expand these findings to elucidate the role of signal detection in IA, especially self-reported accuracy. These mixed findings also highlight the importance of considering how interoception and cognition are measured when designing studies.

### Implicit (*vs* Explicit) Self-Esteem

Two other findings are worth noting. First, although the MAIA-g did not correspond with *signs and symptoms* component scores, it did correspond to greater implicit self-esteem and lower depression scores on the DASS. Second, MAIA-g scores did not correspond to explicit self-esteem yet did correspond to the FSIS and SWLS. Further, despite only correlating independently to the FSIS and SWLS (and not the RSES), the *self-regard* component exhibited significantly better predictive power of the MAIA-g than the IAS, the latter of which correlated with all three questionnaire scores in the component. It is

not clear why this constellation of findings emerged. Regarding self-esteem, one possibility is that the respondents (MTurkers) who completed this survey were especially familiar with the RSES. The RSES is one of the most widely used instruments in crowd-sourced social science research (*Fowler, Jiao & Pitts, 2022*), and social science research in general—a recent systematic review identified 7,760 articles that mentioned or cited the RSES (*Gnambs, Scharl & Schroeders, 2018*). As a result, respondents may have been exposed to the RSES before, potentially multiple times. Since MTurkers (and similar) are incentivized to respond efficiently, they may have responded to the questions with little contemplation (*i.e.,* with acquiescence). The self-esteem IAT is more difficult to fake or engage in acquiescence than the RSES (*Rohner, Schroder-Abe & Schutz, 2011*; *Steffens, 2004*), as is the less-common SWLS and FSIS questionnaires, the latter of which is a semantic differential. The novelty of the measures may have spurred greater engagement. Others have suggested that implicit and explicit self-esteem are two different constructs (*Hofmann et al., 2005*; *Pietschnig et al., 2018*), despite sharing similar names, and it may be that some aspect implicit self-esteem is simply more relevant to subjective interoception. A final consideration is the age-appropriateness of the instrument: some have criticized that the RSES is not an ideal measure for adults,[5] and, as such, may generate ceiling effects (*Butler & Gasson, 2006*).[6] Others, however, argue there is no support for such a claim (*Sinclair et al., 2010*). Nevertheless, researchers would do well to carefully consider the instruments they choose in crowd-sourced data collection.

## Negative affect

Regarding the depression subscale scores, it may be that simply noticing the body's sensations is not sufficient to reduce negative affect. Instead, a broader approach to monitor and understand the body and its reactions—and be accurate in that understanding, as assessed by the IAS—can provide important feedback to an individual who, in turn, is likely to use this information for emotional awareness and regulation (*Trevisan et al., 2019*; *Zamariola et al., 2019*). This information can then result in reduced negative affect and depression risk (*Eggart et al., 2019*). The noticing subscale simply asks about passive awareness of the body whereas the remainder involve comparatively more active engagement with the body (*e.g.,* item 25: "I can reduce my breath to reduce tension," an item on the *self-regulation* scale). Perhaps the contribution of active body monitoring and regulation are more important for well-being than simply noticing one's reactions (*Brand, Petzke & Witthöft, 2022*). However, as with above, we cannot conclude causality based on this relationship as. For example, it is certainly possible that negative affect causes impaired body monitoring and regulation as much as impaired body monitoring can cause negative affect (cf. *Tan et al., 2023*).

## Component structure

The component structure that emerged is also notable. All behavioral measures (flanker, IAT) loaded on to a component with negative affect (depression, anxiety, stress), whereas subjective self-regard (self-esteem, self-image, life satisfaction) loaded onto a second. The first component contributes to growing evidence that affect, (implicit) self-esteem, self-awareness, and cognitive control are related constructs that share similar neurocognitive

[5]The title of the book presenting the original instrument is, after all, "Society and the *adolescent* self-image" (emphasis added).

[6]In the present dataset, there was little evidence of ceiling effects: the mean, median, and modal score was 20 (out of 40).

and neurobiological underpinnings (*Gyurak et al., 2012*; *Pulopulos et al., 2022*). It is not clear why the second component (with all self-report measures of self-regard) diverged from the first. As suggested above, previous work suggests that "objective" (*i.e.,* implicit, symptomatic) and "subjective" measures of self-regard and well-being are weakly aligned and, when analyzed in conjunction with other variables, may not align at all (*Hofmann et al., 2005*; *Pietschnig et al., 2018*). The current component structure may reflect this disparity. These results provide further evidence that self-regard and mental health (including depression/negative affect) are discrete (*Orth & Robins, 2022*), yet correspond to facets of IA.

## Limitations and future directions

Several limitations of this study should be mentioned. First, although *a priori* power analyses indicated this sample size was adequate for the study, others would argue that this sample is not (*Schönbrodt & Perugini, 2013*). These results, therefore, should be viewed as preliminary and requiring further replication, preferably with larger samples. Relatedly, there was an unexpectedly high level of attrition, which is not unheard of in studies that involve relatively lengthy or complex study designs (*e.g.*, *Mancenido et al., 2021*). This attrition is more likely due to inattentiveness than malevolence or "bots," the latter of which would likely have been caught by MTurk's or Qualtrics' built-in algorithms (*Mancenido et al., 2021*). Further, the present measures of interoception were entirely self-report; future work should consider including behavioral assessment of IA to include in analyses. Some have been able to acquire objective IA data remotely (*Morelli et al., 2018*; *Murphy et al., 2020*), which this is an exciting avenue for future work to expand upon these findings and better understand interoception and its associated processes. The use of crowdsourced data presents its own set of restrictions. A motive for using this method was to increase diversity in the sample, yet the composition of the sample was not especially diverse, ultimately limiting generalizability. Future researchers should pursue a greater balance of gender, race, and other demographics. Relatedly, though recruitment was limited to MTurk accounts registered in the United States, it is impossible to verify that participants were, in fact, physically in the United States. Additionally, national origin and cultural background information was not collected. Given potential confounds of language and cultural background that are often tied to national origin, especially in the domain of interoceptive accuracy (*Ma-Kellams, 2014*; *Prentice et al., 2022*), these are all important variables to consider going forward. The use of crowdsourced data also precluded controlling for confounds like environmental distractions. However, it has long been shown that unsupervised and/or crowd sourced measures of cognition correspond well to performance in-lab (*Anwyl-Irvine et al., 2020*; *Cromer et al., 2015*; *Miller et al., 2018*). Nevertheless, future work would do well to replicate and extend these findings in a variety of settings. In terms of the measures themselves, the flanker task is a complex, nuanced task and presented here is a small slice of those data. Expanded analysis of this task is certainly warranted in future work. Further, measurement of attention outside of the visual domain may be helpful to expand the diversity and representation than was in the present sample (*e.g.*, individuals with disabilities). As well, there is an updated

version of the MAIA that is now available (*Mehling et al., 2018*). However, the only changes in the updated version are the addition of five items across the two subscales excluded in the MAIA-g (*Ferentzi et al., 2021*) and from this study. Should these additional items improve the MAIA, especially the MAIA-g and its psychometrics, then it is certainly worth considering using this measure.

## CONCLUSION

In short, monitoring the body and accurately perceiving its signals are both processes that correspond to well-being (with subtle differences) but differentially correspond to cognition, namely selective (visual) attention. Interoception contributes to an overall more positive experience in daily life and greater regard for self and others. Though ample previous work suggests that facets of interoception corresponds to self-referential processing, this work provides the first evidence that the "self" involved in those processes is a valued entity. These results also suggest that subjective interoceptive attention, subjective interoceptive accuracy, exteroceptive attentional abilities, self-regard, and well-being may operate in unique, but related, dimensions. Depending on the measure used, interoceptive attention also corresponded to self-regard and well-being to a greater degree than subjective accuracy, yet interoceptive attention negligibly corresponded to visual attention. Interoceptive sensibility may be a fruitful avenue for future work pertaining to cognition and emotion, and greater monitoring and variety of instruments and methods to assess these constructs should be employed to elucidate these associations further.

### Funding

The author received no external funding for this work.

### Competing Interests

The author declares that they have no competing interests.

### Author Contributions

- Erik M. Benau conceived and designed the experiments, performed the experiments, analyzed the data, prepared figures and/or tables, authored or reviewed drafts of the article, and approved the final draft.

### Human Ethics

The following information was supplied relating to ethical approvals (i.e., approving body and any reference numbers):

The Institutional Review Board at SUNY Old Westbury approved the procedures in the current study.

### Data Availability

The data are available at Open Science Foundation (OSF): Benau, Erik. 2023. "sIA, Attention, and Self-Regard." OSF. January 17. doi: 10.17605/OSF.IO/7SXE3.

## Supplemental Information

Supplemental information for this article can be found online at http://dx.doi.org/10.7717/peerj.15348#supplemental-information.

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
