# Peer review of "Self-reported interoceptive accuracy and interoceptive attention differentially correspond to measures of visual attention and self-regard"

_PeerJ, doi:10.7717/peerj.15348_

## Round 0.1 · original submission · Major Revisions

Dear Dr. Benau,

Thank you for submitting your manuscript to PeerJ. Your paper was evaluated by three individuals with considerable expertise in this area. One of the reviewer is also an expert in quantitative psychology. All three reviewers have found your research question and study very interesting and have commented on a number of significant strengths. At the same time, they have identified several concerns. Based on my independent reading, I agree that your paper has potential for an impactful contribution, but some revisions are necessary. Thus, I am inviting you the option of revising and resubmitting it for consideration.

The reviewers have provided exceptionally thorough and constructive comments. I will not repeat their points but I urge you to attend to all of their points, especially those related to the validity of the findings and your statistical analyses. You are welcome to submit a revision, if you think that you can address all concerns.

Thank you for giving us the opportunity to consider your work, and we wish you all the best with your article.

Yours sincerely,

Andree Hartanto
Academic Editor
PeerJ

Reviewer 1 ·

Basic reporting

The paper was relatively clear and easy to understand, and the author’s dedication to transparency and openness is commendable. Nonetheless, it would be beneficial for the author to reduce the use of acronyms to improve the readability of the paper. Specifically, the acronyms “sIAcc” and “sIAtt” were not particularly intuitive, especially given that these constructs were measured using the “IAS” and “MAIA” (and “MAIA-g” was also involved). It was slightly difficult to follow the paper’s arguments at some points of the work given the many acronyms used in the paper.
Additionally, it was not particularly evident that the current work focuses on visual attention (as mentioned in the title), given that PCA analyses grouped visual attention measures with negative affect. Hence, the title appeared slightly misleading, given the relatively sparse discussion on the association between sIAcc and sIAtt with visual attention specifically.
It would also be beneficial for the author to elaborate more on why there might be a difference regarding sIAcc and sIAtt in the introduction of the paper. While the present work is undoubtedly novel, the work will be greatly strengthened by a more detailed theoretical elaboration on why the author would expect sIAcc and sIAtt to be differentially associated with self-regard and well-being. Although the author has made an attempt to do so in Lines 93 to 102, the work would be greatly strengthened with further elaboration on how self-introspection accuracy and attention differs, and why they might be differentially associated with the factors examined in the paper.

Experimental design

The goals of the study was clearly stated (Lines 119-121). The methods and measures were described with sufficient detail.

Validity of the findings

One particularly relevant limitation to the current work is with regard to the IAS measure as a measure of “accuracy”. Specifically, the IAS appears to measure participants’ perceived accuracy more so than an objective measure of accuracy. This raises the concern that there is a confounding variable of positive self-perception. More specifically, might individuals who perceive themselves to be more accurate, also perceive themselves to be better (and thus have higher self-regard)? Should this be the case, individuals’ actual (rather than self-perceived) accuracy may be unassociated with self-regard. A discussion of this point would greatly strengthen the present work. The current lack of discussion regarding this point also undermines the current theorizing that actual accuracy (e.g., an objective measure like predicting sneezes, as highlighted in Line 85) is what is associated with self-regard and well-being.

Additional comments

There are some additional minor comments that the author should address:
a. There is likely a typo in Line 157 or 158. The author states that the FSIS was measured on a seven-point scale (Line 157). However, the anchors of the scale (Line 158) show that the scale was an eight-point scale ranging from 0 to 7.
b. It would be beneficial for the author to elaborate on the use of PCA. Were there apriori predictions regarding how the variables assessed (and subjected to the PCA) were related? Additionally, the author states that the PCA was used to reduce the number of variables examined. This appears to differ from the goals of the study stated in Lines 119-121, which appears to suggest that the PCA was done to establish whether indices of self-regard, well-being, and visual attention correspond to each other. What was the main aim of the PCA?

Overall, the work has great potential in contributing novel insights on the relationship between interoception and self-regard. The work is also commendable in providing a thorough summary of the analyses conducted and the respective findings from these analyses. The work can be strengthened with more theoretical discussions on the motivation behind the current work.

Reviewer 2 ·

Basic reporting

I thank the Author and Editor for the interesting read. I find the core topic of examination (association between interoception facets and well-being) to be quite an interesting one, with potential clinical implications through further studies. There are, however, a number of concerns that precludes me from being able to recommend the manuscript, as it is, for publication (as reported within the respective sections' fields here).

Broadly speaking, for the introduction, while noticeable groundwork has been put in place to introduce the key variables, the theoretical rationale linking the suite of variables discussed and well-being remains quite unclear. I note that DV-focused discussions within the introduction had centred almost exclusively around self-regard, and, even for self-regard, I felt that the underpinning theoretical rationale could have been better structured and fleshed out.

A relatively minor point – certain areas of expressions and presentation could have been further improved upon for clarity and flow. For instance, the abstract appears to be a collage of lists and, as a result, reads a little disjointed. The use of acronyms was not very consistent as well. For example, on page 10, it was stated that a goal was to “determine whether these measures correspond to sIAcc and/or (c) interoceptive attention.” even though “interoceptive attention” had already been pre-defined as sIAtt.

Experimental design

Hypotheses development and substantiation, I felt, could have benefitted from further elaborations and perhaps headers/sub-headers. It is quite unclear why the Author expects and seeks to examine an intercorrelation between self-regard, well-being, and visual attention (i.e., “goal (a)”) – with visual attention being the least apparent on why it is grouped together with the other two variables.

From what I can garner, the Author may have been trying to propound the abovementioned variables as a collective DV (or Outcome variable), while “goal (b)” might have then been the IVs (or predictors). Should that be the case, I would recommend revisions in structuring and presentation to better convey this. But, even in this scenario, it still remains nebulous as to why “visual attention” was included alongside the other two variables here.

Relatedly, I found the conducting of PCA to be somewhat puzzling. If the goal here is to produce a composite index among those variables, then a) a strong and clear theoretical rationale is needed to explain why those variables can and should be truncated together and what its resultant index score mean, and b) structural equation modelling (SEM) could have been considered as a more rigorous (and elegant) approach.

Validity of the findings

More details on how the a priori power analysis was conducted would have been beneficial (e.g., type/family of power analysis, referenced effect size, etc.).

Given the, arguably, alarmingly large attrition rate, the Author could consider conducting a missing data analysis (e.g., are the data missing completely at random, etc.)

It might be helpful for prospective readers to at least duly acknowledge the ongoing controversies surrounding the validity of the IAT and its derivatives (e.g., https://www.ncbi.nlm.nih.gov/pmc/articles/PMC8167921/)

Additional comments

none

Reviewer 3 ·

Basic reporting

Please reduce the number of acronyms/abbreviations used throughout the manuscript, other than those that are well-known across all fields (e.g., PCA).

Please justify why “It was hypothesized that measures of negative affect, self-regard, and attention would form a unitary construct”. There was no explanation of this in the introduction.

Experimental design

The plan of analysis for the multivariate regression portion is lacking. Please provide more information in the methods. Ideally, explicit mathematical equations could be provided for maximum clarity. If not, please phrase the model clearly.

The word “unrotated” in “unrotated PCA” (line 259) is redundant; PCA is always unrotated by definition. It is unclear why rotation was attempted to “clarify interpretation” (and in any case, the purpose of PCA is not to derive interpretable factors in the first place); either remove this or explain further. Alternatively, I wonder if the author instead conducted EFA (where rotation and interpretable factors do come into play) but misstated it as EFA. Please clarify.

Correlations are usually not stable at sample sizes below 250 (Schönbrodt & Perugini, 2013, https://doi.org/10.1016/j.jrp.2013.05.009). I would *strongly* encourage the author to continue collecting more data via MTurk until at least N=250 is achieved. If this is not possible, then this limitation should be explicitly mentioned in both the methods and in the discussion, and this work should be framed as preliminary in the introduction and discussion.

Validity of the findings

No comment

Additional comments

I have provided extensive comments (incl. those covered above) in the attached document. I hope the author finds them useful and takes them into account.

Annotated reviews are not available for download in order to protect the identity of reviewers who chose to remain anonymous.

---

## Round 0.2 · Minor Revisions

Dear Dr. Benau,

Thank you for submitting your manuscript to PeerJ. We have just obtained the reviews from our experts. I also have read the manuscript myself independently before looking at the reviews. Overall, we are satisfied with the revision and agree that your paper has potential for an impactful contribution.

The reviewers also raised a number of relatively minor concerns that should be easy for you to address. Pending the minor revision, I am happy to conditionally accept your paper for publication in PeerJ

Thank you for giving us the opportunity to consider your work, and we wish you all the best with your article.

Yours sincerely,

Andree Hartanto
Academic Editor
PeerJ

Reviewer 1 ·

Basic reporting

The revised manuscript is substantially stronger than before, and the framing of the paper is much clearer now. From the revised manuscript, it is much clearer that exteroceptive attention (visual attention) is an integral and important part of the paper. The reduced number of acronyms have made the paper much easier to follow as well. I appreciate the author’s receptiveness to the reviewers’ comments, and think that the author has adequately addressed my previous comments.

I apologize for not bringing this up previously. However, I believe that a results table summarizing the key findings presented in the subsection “Relation of principal component scores to self-reported interoceptive accuracy and interoceptive attention” will greatly benefit readers in allowing them to view the study’s findings at a glance.

Experimental design

I have no further comments on the experimental design.

Validity of the findings

I have no further comments on the validity of the findings, and appreciate the effort the author has put into refining the limitations and introduction sentences.

Additional comments

Again, I appreciate the author’s meticulous work in revising the manuscript. I have no further suggestions and concerns besides the suggestion regarding the inclusion of a results table.

Reviewer 3 ·

Basic reporting

Line 236, is "discreet" (careful/confidential) supposed to be "discrete" (separate)?

For ease of reading, the paragraph from line 684 to line 706 can be split into two paragraphs at line 690 ("their income. // There is no established protocol ...").

Experimental design

No comment

Validity of the findings

No comment

Additional comments

I am overall satisfied with the revisions made by the author. Minor improvements could be made to make the writing more succinct throughout the introduction and discussion, but the logic and arguments made are fine. Thank you for this interesting piece of work.

---

## Round 0.3 · accepted · Accept

Dear Dr. Benau,

I am pleased to advise that the above paper has now been accepted for publication in PeerJ. Thank you for giving the Journal the opportunity to publish your work. We are impressed with your paper and believe that it will contribute well to the literature. Well done!

Best Regards,
Andree